# Radiation Treatment Mechanisms of Cardiotoxicity: A Systematic Review

**DOI:** 10.3390/ijms24076272

**Published:** 2023-03-27

**Authors:** Konstantinos C. Siaravas, Christos S. Katsouras, Chrissa Sioka

**Affiliations:** 1Second Department of Cardiology, University Hospital of Ioannina, University Campus, 45110 Ioannina, Greece; 2Department of Nuclear Medicine, University Hospital of Ioannina, University Campus, 45110 Ioannina, Greece; 3Faculty of Medicine, School of Health Sciences, University of Ioannina, University Campus, 45110 Ioannina, Greece

**Keywords:** cardiac toxicities, radiation treatment, cancer

## Abstract

Radiotherapy may be used alone or in combination with chemotherapy for cancer treatment. There are many mechanisms of radiation treatment exposure to toxicities. Our aim was to summarize the literature about known mechanisms of radiation-induced cardiac toxicities. We performed a systematic review of the literature on the PubMed database until October 2022 about cardiovascular toxicities and radiation therapy exposure. Only systematic reviews, meta-analyses, and reviews were selected. Out of 1429 publications screened, 43 papers met inclusion criteria and were selected for the umbrella review process. Microvascular and macrovascular complications could lead to adverse cardiac effects. Many radiotherapy-associated risk factors were responsible, such as the site of radiation treatment, beam proximity to heart tissues, total dosage, the number of radiotherapy sessions, adjuvant chemotherapeutic agents used, and patient traditional cardiovascular risk factors, patient age, and gender. Moreover, important dosage cutoff values could increase the incidence of cardiac toxicities. Finally, the time from radiation exposure to cardiac side effects was assessed. Our report highlighted mechanisms, radiation dosage values, and the timeline of cardiovascular toxicities after radiation therapy. All of the above may be used for the assessment of cardiovascular risk factors and the development of screening programs for cancer patients.

## 1. Introduction

Cancer and cardiovascular diseases are common comorbidities [1]. Chemotherapeutic agents, as well as radiation therapy used for cancer treatment, can cause cardiovascular toxicities [2]. Radiation treatment may be responsible for coronary artery disease (CAD), valvular heart disease (VHD), heart failure (HF), myocarditis, cardiomyopathies, arrhythmias, and pericardial syndromes (acute and chronic) [3]. There are many mechanisms responsible for radiation treatment-induced cardiotoxicities [4].

Radiation therapy is a common adjuvant cancer therapy to chemotherapy or surgery for many common types of cancer. Breast, lung, and esophageal cancer and lymphomas may expose the heart to the deleterious effects of irradiation [5,6]. There are several literature reviews suggesting possible mechanisms of cardiac toxicities of radiation therapy and treatment options [7].

Another important aspect is the timeline between exposure to radiation therapy and the appearance of cardiac toxicities, as well as the radiation dosage to which the heart is exposed. Many cardiac toxicities can emerge acutely after initiation of radiotherapies, such as pericardial effusion or acute pericarditis, arrhythmic events, and conduction abnormalities, while others, such as coronary artery disease, valvular heart disease, chronic pericardial syndromes, and constriction may appear many years after exposure.

In the present review, our aim was to summarize all possible mechanisms of radiation-induced cardiotoxicity suggested in the literature. Furthermore, we will present information on irradiation-associated risk factors for cardiovascular disease and time from radiation exposure to cardiac toxicity appearance.

## 2. Methods

### 2.1. Search Strategies and Selection of Studies

The PRISMA statement for reporting systematic reviews was used for methodological guidance in the current report. The PubMed database was searched until October 2022. The following search terms were used: radiation treatment OR radiation therapy AND cardiotoxicity OR cardiac toxicity. Filters for systematic review, review, and meta-analyses were used for article selection.

The following inclusion criteria were applied for paper selection during the review process: there were selected only literature papers with a format of review, systematic review, and meta-analyses reports, which were published until October 2022, that include information about mechanisms of radiation treatment cardiotoxicities, time from radiation exposure to cardiac toxicity and total radiation dosage and radiation associated risk factors for cardiovascular disease. Only reports published in the English language were selected due to language barriers. Figure 1 presents the flowchart of paper selection.

### 2.2. Literature Search and Study Selection

We reviewed a total of 1429 publications and excluded all publications with irrelevant titles and abstracts. All papers which were not reviews, systematic reviews, or meta-analyses were also excluded, and the remaining 99 publications were further assessed. Forty-nine reports without data about mechanisms of radiation treatment cardiotoxicities were further excluded. After the initial assessment, 50 publications were included for further analysis, meeting the eligibility criteria. There were no duplications in the papers to be excluded. Seven reports which were not in the English language were also excluded. Finally, 43 papers were selected for inclusion in the current review.

## 3. Results and Discussion

### 3.1. Mechanisms of Cardiotoxicity

There are many reported mechanisms of cardiotoxicity caused by radiation therapy exposure in cancer patients. Cardiac cell dysfunction due to oxidative stress from radiation exposure is an important mechanism of cardiac toxicity. Mitochondrial dysfunction due to radiation can break the respiratory chain of metabolism, causing an increase in the production of Reactive Oxygen Species (ROS). Antioxidant capacity cannot override this increase, leading to mitochondrial instability and dysfunction. ROS can cause macromolecule (lipid, protein, amino acid, etc.) damage. Apart from increased ROS formation, direct DNA damage from radiation leads to DNA strand breaks, causing genome instability. Impaired DNA correction mechanisms are not working properly under radiation exposure, becoming unable to correct the apparent DNA damage, leading to the activation of apoptotic mechanisms [8].

Radiation leads to macromolecule damage causing cardiac cell damage. Protein oxidation under increased ROS production cause amino acid chain breaks, misfolding and cross-linking, and dysfunction of degrading protease proteins. Many cytocellular metabolic mechanisms are affected because proteins function in many different cellular components, e.g., as receptors, enzymes, and transporters. Furthermore, ROS induces lipid peroxidation causing polyunsaturated fatty acid (PUFA) to form fatty acid radicals. Many antioxidative and other cellular membrane mechanisms can be deactivated by this transformation. All the above radiation-induced mechanisms cause molecular signaling pathway damage and cause cardiac toxicity [8].

Oxidative stress increases TGF-β1 leading to radiation-induced vascular injury and endothelial dysfunction. Furthermore, NF-kB upstream tumor necrosis factor (TNF) and interleukin (IL-1) production create a pro-inflammatory environment for cardiac cell inflammation and dysfunction [8]. All the aforementioned mechanisms can activate cellular apoptosis.

Furthermore, even 8 Gy of radiation dosage can stimulate the release of calcium from the endoplasmic reticulum to the cytoplasm leading to an increase in cytoplasmic calcium concentration. The above mechanism can increase mitochondrial ROS production and activate p53 promoting the accumulation of pro-inflammatory and prothrombotic molecules and increasing the probability of thrombosis, myocardial ischemia, and inflammation. Radiotherapy can be an alternative factor of atherosclerosis through capillary rupture and microvascular damage [9].

Non-coding RNAs (ncRNAs) are genetic, epigenetic, and translational regulators. NcRNAs mainly compose of long non-coding RNAs (lnc-RNAs), microRNAs (miRNAs), and circular RNAs (circRNAs). NcRNAs play an important role in apoptotic mechanisms (Fas pathway and caspase pathway cascade), mitochondrial damage (detoxification mechanisms disruption), oxidative stress (increase ROS production), autophagy, calcium homeostasis, and fibrosis. Radiation-induced miR-34a promotes gene aging and decreases the length and activity of nucleic acid telomeres. All the above induce cardiomyocyte aging and disruption. Some miRNAs can be used as a biomarkers for the early detection of cardiac cell toxicity after radiation therapy and early detection of CVD but they also can be used as potential therapeutic targets to reverse these toxic effects [10].

In addition, the following mechanisms cause CVD after radiation therapy. Interstitial fibrosis cause subsequent diastolic dysfunction left ventricular dysfunction, or left ventricular remodeling, and fibrosis of the conduction system may provoke arrhythmias or conduction defects. Pericardial fibrosis is important for many pericardial syndromes [11].

Furthermore, radiation exposure cause endothelial changes, which result in the reduced capillary-to-myocyte ratio, damage to epicardial vessels, and disrupt the balance between antithrombotic and prothrombotic, as well as anti-inflammatory and pro-inflammatory mechanisms. Induced monocyte and macrophage accumulation to the atherosclerotic plaque as well as enhancement of the action of classic cardiovascular risk factors, increase and accelerate the process of atherosclerosis in cancer patients. Studies in murine models after irradiation demonstrate structural changes in atherosclerotic plaques leading to more intraplaque hemorrhages and plaque instability that is vulnerable to thrombosis [12].

Furthermore, pathophysiological changes of radiation treatment exposure and induced cardiac toxicities have been studied in animal models, and information about fractionation of dosage and radiation protection treatments have been tested [13].

Deposition of extracellular matrix and fibrosis deteriorates the elastic structure of vessels, resulting in arterial rigidity. Many inflammatory molecules and cytokines play a detrimental role in this process, such as tumor necrosis factor (TNF), interleukin (IL)-1, IL-6, platelet-derived growth factor (PDGF), monocyte chemotactic factor (MCF), and transforming growth factor (TGF)-b. Endothelial damage from radiotherapy disrupts endothelial-derived production of vasodilator factors, such as nitric oxide (NO). The consequent vasoconstriction and the activation of the prothrombotic environment can lead to hypoperfusion and microvascular damage [14]. Histological studies report two main complications on arteries: intimal disruption and hyperplasia and luminal stenosis or occlusions [15,16].

Another complication of radiation therapy consists of autonomic dysfunction. It may induce abnormal heart rate recovery time and increased resting heart rate [12]. Clinical radiotracer studies suggest that myocardial alterations in metabolic uptake are present in irradiated heart tissues, which are not linked to vascular territories of coronary arteries. Metabolic shifts may occur due to switching from fatty acid oxidation to glycolysis, eventually leading to increased glucose uptake, mitochondrial damage and radiation-induced heart disease (RIHD), and ischemic heart failure [17]. Zebrafish models are used for the characterization of the pathophysiology of acute, subacute, and delayed RIHD and for the identification of potential cellular and molecular therapeutic targets [18]. Figure 2 summarizes different mechanisms of radiation-induced toxicities, and Table 1 summarizes the clinical consequences and their associations with pathophysiologic mechanisms of radiation-associated cardiotoxicities.

### 3.2. Cardiac Toxicities Associated with Radiation Exposure

Coronary artery disease (CAD) after radiotherapy is a consequence of endothelial damage, which leads to a pro-inflammatory state and, via oxidative stress, disrupts DNA strand integrity accelerating atherosclerosis pathophysiology. Proximal coronary artery lesions and ostial lesions incidence is increased in cancer patients treated with radiation therapy. A paper reported that cancer patients usually had proximal lesions, and the most commonly affected vessels were the left anterior descending artery (LAD) in about 49% and the right coronary artery in 33% of patients. Optical coherence tomography findings in culprit lesions underlined that cancer patients had a statistically significant increased incidence of plaque erosions and calcified nodule culprit lesions, whereas non-cancer patients had an increased incidence of plaque rupture. In addition, cancer patients had statistically more calcifications of culprit lesions versus more prevalence of thin cup fibroatheromas plaques of non-cancer patients [19]. The timeline of coronary artery disease after radiation treatment might start from 5 years after exposure or even after 20 years. Patients with RIHD seem to have greater mortality after percutaneous coronary interventions than patients without a radiation treatment history. Even patients revascularized with coronary artery bypass grafting have worse outcomes versus control patients without cancer history [20].

Valvular heart disease can be a sequence of valve tissue fibrotic changes and thickening, with or without calcifications. The most commonly affected heart valves are the aortic and mitral. Left-sided heart valves are more prone to irradiation exposure due to their anatomic location and proximity to irradiated tissues. CAD and VHD may have a slow progression in time and might take several years, even after cessation of radiation therapy, to evolve [18,20].

Radiation significantly increases the risk of non-ischemic cardiomyopathies. The pro-inflammatory state can result in differentiation of muscle cells into myofibroblasts through inflammatory cytokines excreted by inflammatory cells, leading to collagen accumulation and fibrosis, causing heart tissue remodeling. All of the above can cause diastolic dysfunction and stiffening of the affected myocardium. Another factor of increased prevalence of myocardiopathy is the fact that other cardiac toxicities can also affect heart tissue remodeling in a way that increases the risk of cardiomyopathies, e.g., hypertrophic remodeling due to concomitant VHD or restrictive cardiomyopathy and constrictive pericarditis [21].

The chronic inflammatory process may be sustained after the received radiation treatment exposure and may be responsible for acute and chronic pericardial syndromes. Acute pericarditis may have been reduced in recent years due to advancements in radiation therapy techniques, but chronic pericardial effusions and pericardiac constriction have increased. The chronic pericardial inflammatory process can cause thickening, increased rigidity, calcification of the pericardial sac, and loss of pericardial compliance [22]. Pericardial thickening, neutrophil infiltrations, and calcifications can cause constriction pathophysiology [23].

Direct damage of irradiation to conduction system cells and indirect inflammatory response to irradiation and conduction system fibrosis are responsible for arrhythmias in cancer patients. Right bundle branch block (RBBB) is the most common conduction abnormality among irradiated cancer patients and can appear even after years to decades after treatment cessation [24]. Other vascular complications of irradiation of cardiovascular tissues are carotid and peripheral vascular disease, through direct irradiation damage or similar pathophysiologic mechanisms to accelerated atherosclerosis and coronary artery disease.

Cancer and its treatments are important factors for arterial and venous thromboembolic events, such as venous thromboembolic disease, pulmonary embolism, and arterial thromboembolic events. Until recently, there were not enough data for the use of nonvitamin k oral anticoagulants (NOACs) in cancer patients. However, nowadays, with all the novel evidence, there is a growing trend of using NOACs as the preferred anticoagulant for atrial fibrillation and thromboembolic events. It is useful to underline the fact that metabolic pathways of catabolism of NOACs (through cytochrome P450 3A4) can cause drug interactions with common chemotherapeutic agents used in cancer treatment (antimetabolites, topoisomerase inhibitors, anthracyclines, alkylating agents, tyrosine kinase inhibitors, etc.) and thus it is imperative a careful monitoring of side effects and possibly NOAC dose reduction and adjustment according to these interactions [25]. Table 2. summarizes the effects of radiation and chemotherapeutic agents on cardiac toxicities after cancer treatment.

### 3.3. Risk Factors for Radiation-Induced Cardiotoxicity

There are many risk factors deteriorating cardiotoxic effects of radiotherapy, such as cumulative radiation dose, irradiated heart volume, young age at the time of radiation exposure (especially doses >30–50 Gy), fractional dosage (>2 Gy), heart volume irradiated, young patient age at the time of radiation exposure, the time elapsed since exposure, concomitant use of other chemotherapeutic agents and classical cardiovascular risk factors presentation [11]. Multiple factors, including cellular composition, differentiation, cell replication capacity, and cellular radiation sensitivity, determine the extent of possible toxicities [26]. The adult heart is considered unable to regenerate after fully developed and thus has been viewed as a radioresistant organ [27].

Higher doses of radiation treatment accelerate cardiovascular toxicities. Much effort has been made with newer technologies to reduce the total therapeutic dose needed and reduce the heart volume irradiated. There is a linear increase in cardiac toxicity per Gray of radiation dose increase [28]. Prior 2D-based radiation therapy was replaced by more conformal approaches such as 3D conformal radiation treatment (RT)and proton beam therapy [29]. Recent high-precision treatments such as intensity-modulated radiotherapy (IMRT), stereotactic radiotherapy (SRT), and image-guided radiotherapy (IGRT) improved quality and suppressed the cardiotoxic effects [30].

### 3.4. Radiation Dosage for Cardiac Toxicity

Different radiation therapy dosage cutoffs might be responsible for cardiac toxicity according to cancer type. Furthermore, accumulating dosage and proximity to heart tissue are important factors for the radiation dosage quantity needed for cardiac toxicities to exist. Additionally, lower radiation dosage for long periods of time or lower dosages that lead to a more chronic and late (after many years from exposure) are also important for adverse cardiac effects.

A dosage of 40–44 Gy for patients treated for Hodgkin’s lymphoma might be detrimental to the development of toxicities. Radiation dosage > 30 Gy may be enough to create adverse cardiac toxicities, and in combination with chemotherapy, can further reduce cutoff radiation dosage to 20 Gy. With the advancement in techniques of radiation treatment, a mean heart dose cutoff of >25 Gy may be used for adverse cardiovascular effects. An important aspect is that with newer advancements, a mean heart dose < 10 Gy may be enough to treat cancer patients, thus minimizing radiation exposure to heart tissues. Pediatric lymphoma patients potentially develop heart failure events after 15 Gy of radiation, whereas myocardial infarction occurs after 35 Gy [29].

In breast cancer patients, an increase in mean heart dose of 5 Gy may significantly increase the deleterious effects on heart tissues. For esophageal cancer patients, 40 Gy of radiation therapy might cause radiation-induced heart disease. For lung cancer, the volume of heart tissue radiated at 40 Gy is important for possible side effects [12].

### 3.5. Time from Radiation Therapy to Cardiac Side Effects

Radiotherapy can cause radiation-induced heart disease. There is a different timeline from radiation treatment exposure until the cardiotoxic effects appear. Additionally, every distinct cardiac toxicity (CAD, VHD, cardiomyopathies, pericardial syndromes, conduction abnormalities) needs a different time from exposure to appear.

Coronary artery disease is the most common clinical presentation after radiation therapy exposure. Even 15 years after mediastinal irradiation is enough to cause CAD. The majority of CAD cases after radiation treatment are seen between 10 to 30 years. A greater incidence is expected, especially in pediatric cancer survivors, since there is a younger exposure and more time until the clinical presentation of the disease. Furthermore, there is more time for simultaneous classic cardiovascular risk factors to act additionally to radiation exposure. Those patients can present as asymptomatic coronary stenosis as, acute coronary syndromes, or event as a silent infarction. The distribution of coronary arteries affected (left or right coronary vascular system) is influenced by the site of radiation beam exposure. Furthermore, cancer survivors present with a greater extent and severity of CAD than non-cancer patients [21].

The mean interval of valvular heart disease is 23 years after radiation exposure. One-third of cancer patients face VHD after 10 years, whereas the majority of cases, and around 60% of patients, after 20 years. The most commonly affected valves are the aortic and mitral. With all technological evolvements nowadays, more cancer patients with the valvular disease after radiation exposure are managed on an interventional, and apart from classic surgical options, now many cases of transcatheter valve replacement take place.

Cardiomyopathy incidence increases after radiation exposure. Non-ischemic cardiomyopathies, due to the direct effects of radiation and interstitial fibrosis, were observed in cancer patients. A shorter timelapse of around 3–4 years median time from radiation is needed for these clinical toxicities to appear. Most cases are clinically with heart failure symptoms.

Pericardial disease (acute pericarditis, pericardial effusion, delayed thickening, and constrictive pericarditis) occurs event sorter with a mean time from exposure of 6 months. Acute pericarditis may develop days or weeks after exposure, but pericardial effusions may develop after weeks or months. Constrictive pericarditis may have a latent period of up to 10 years after the radiation therapy develops [31]. Thus, pericardial diseases have dual time peaks for development after radiation treatment exposure, with an acute peak of acute pericarditis after weeks or months of exposure and a later peak after years for pericardial delayed thickening and constrictive pericarditis. Furthermore, conduction abnormalities occur after 2 months of completion of therapies [32]. Figure 3 shows the time course of cardiac toxicities after radiation treatment.

### 3.6. Imaging for Detection of Cardiovascular Toxicities

Multimodality cardiovascular imaging can be very useful in the early detection of cardiovascular toxicities. Echocardiography is the first-line imaging modality used in most cases since it has no adverse effects and is widely available. It can be used to diagnose ventricular systolic or diastolic dysfunction, as well as valvular heart disease or pericardial syndromes. Left ventricular ejection fraction (LVEF) has diagnostic and prognostic meaning for cancer patients, and with advancement, in 3D echocardiography, there is a more precise estimation of its value. Strain echocardiographic imaging is useful for the detection of preclinical adverse effects even before structural heart disease exists [5].

Some important changes that can be detected on echocardiography are regional wall motion abnormalities and reduced left ventricular systolic function, especially in patients with coronary artery disease. Heart failure and cardiomyopathies may present either as reduced systolic left ventricular function (a consequence of CAD or direct toxic effects of irradiation) or may present with preserved systolic left ventricular function as myocardial hypertrophy, diastolic left ventricular dysfunction, and restrictive physiology. Valvular heart disease shows characteristic valvular abnormalities with calcification or degenerative changes leading to valve regurgitation or stenosis and multivalvular heart disease. Pericardial syndromes may be present as pericardial effusion or tamponade findings [33].

A recently published paper from ESC (European Society of Cardiology) summarizes the guidelines on the management of patients with cancer history and cardiac toxicities and specifies initial assessment and calculation of risk, follow-up visits, and biomarker schedule of follow-up and preventive and therapeutic strategies [34].

Cardiovascular computed tomography can be used to calculate scores for the calcification of valves or coronary arteries and for the estimation of valvular and cardiac function. CT coronary angiography can be used for diagnostic purposes to detect coronary artery disease and vessel obstruction. Additionally, is a useful modality also for vascular disease estimation, such as aneurysms.

Cardiac magnetic resonance (CMR) and its advancements are very helpful in diagnosing many cardiovascular toxicities, from differential diagnosis of heart failure and cardiomyopathies to valvular function information and estimation of the severity of the disease. Magnetic resonance can add important diagnostic and prognostic information about cardiac function and, in conjunction with echocardiographic and CT findings, can further help in the early detection of cardiac irradiation damage. Additionally, CMR may provide information concerning left ventricular function and ejection fraction, and with stress MRI techniques, important implications for ischemia assessment in patients with CAD. Furthermore, CMR, in combination with echo findings, can assist in the quantification of valvular heart disease, assessment of myocardial fibrosis, or severity of pericardial syndromes [29]. Nuclear cardiology is a powerful diagnostic tool helping everyday clinical practice detect of toxic effects of radiation treatment. SPECT and PET are very useful for the detection of coronary artery disease, but also heart failure and pericardial syndrome cases [5].

Coronary artery lesions and characteristic plaque findings have further been evaluated with newer intravascular coronary imaging techniques, such as intravascular imaging ultrasound (IVUS) and optical coherence tomography (OCT). By utilizing this technology, it has been able to obtain important measurements of stenotic lesions, minimal luminal area (MLA), and characteristics of atherosclerotic plaques, but also calcification and thrombus formation in cancer patients with coronary artery disease and may guide more appropriate therapeutic management. After percutaneous coronary interventions, further data acquisitions on stent apposition and follow-up studies of in-stent restenosis add important improvements to advanced coronary interventions since lesions of coronary patients may be complex and advanced interventional techniques may be used for therapeutic management (rotablation techniques, shock wave balloons, intravascular lithotripsy) [35]. OCT may be additionally a very useful imaging modality for treatment decisions on dual antiplatelet therapy discontinuation after percutaneous coronary interventions [36].

### 3.7. Strategies for Prevention of Radiation-Induced Cardiac Toxicities

Nowadays, no specific prevention and treatment strategies exist for radiation-induced cardiac toxicities. Classic cardiovascular cardiac risk factors need to be managed according to present guidelines. Most advances have been made in radioprotective oncology leading to minimization of mean heart dose, more targeted application of dosage, and gating techniques (intensity-modulated radiotherapy), which tend to reduce heart tissue irradiated. According to the cardiovascular risk of cancer patients, it is proposed that prevention therapies should be considered for patients receiving certain categories of chemotherapeutic agents, which may also be cardiotoxic. Prevention strategies with angiotensin-converting agents and beta-blockers, as well as statins, may reduce the toxic effects of cancer treatments [34].

### 3.8. Discussion

There are many pathogenic mechanisms aforementioned which are responsible for cardiovascular toxicities of radiation therapy exposure. Many technical characteristics and strategies of radiation therapy, as well as some specific risk factors (patient risk factors, classical cardiovascular risk factors, as well as risk factors and technical considerations associated with radiation treatment therapy on its own), can be responsible for the cardiac toxicities to exist [37].

Due to the great importance of research according to mechanisms of cardiovascular toxic effects of radiation treatment, many advancements have been made in animal models used to configure toxic radiation exposure effects and study pathophysiologic mechanisms [38].

Most interest is given to left ventricular function and pathologies of cardiovascular toxicities. Even though important and accelerated right ventricular remodeling after irradiation is responsible for many clinical syndromes. Right ventricular free wall abnormalities or decreased right ventricular systolic function are common abnormalities after radiation exposure. The right ventricle and its orientation within the thoracic cavity make it more susceptible to deleterious effects of radiation exposure. Right ventricular deformation can play an important prognostic role for cancer patients. Risk factors and technical aspects of radiation therapy, as well as radiation dose and site of implication, are similar to factors that influence left ventricular pathological effects [39].

There is great usefulness in the knowledge of mechanisms of radiation-associated cardiac toxicities. Many pathological and biological pathways may be used as a treatment target for reversing toxic effects, and many molecules can be used as biomarkers of cardiac dysfunction. Many exploratory prevention treatments for RIHD, such as aspirin, statins, and colchicine, have been tested. Due to the anti-inflammatory properties of such drug regimens there might be a reverse of pro-inflammatory mechanisms of radiation-induced toxicities in cancer patients. Aspirin is not used for primary prevention of vascular cardiotoxic effects because no treatment benefit outweighs possible adverse usage effects (such as increased bleeding risk). There is a promising use of aspirin in cancer populations because of the reduction in radiation-induced oxidative damage to tissues that are assumed in many animal models used to test the effects of aspirin in cancer populations. In addition, statins might also have a cardioprotective role in cancer patients because they can diminish endothelial damage and its consequences on radiation-induced cardiac toxicities. Furthermore, atorvastatin can reduce TGF-β adverse effects on capsular fibrosis in rat models. Last but not least, colchicine has potent anti-inflammatory properties due to a reduction in levels of IL-6 and leukocyte adhesion molecules that are responsible for chronic inflammatory and pro-inflammatory properties of radiotoxicity [40].

Serum biomarkers and strain rate imaging can be used for the early detection of cardiac impairment. Brain natriuretic peptide (BNP) may be used as a potent biomarker for early detection of cardiovascular toxicities after radiation treatment initiation, as well as for the prognostic impact on cancer patients. A meta-analysis of breast cancer patients with radiation exposure history showed a significant benefit of the use of BNP as a biomarker for early detection and follow-up of patients for cardiotoxicities [41,42]. Furthermore, strain rate imaging can be used for the early detection of myocardial damage. The higher prognostic benefit is detected in strain left ventricular imaging and is even greater than simply calculating left ventricular ejection fraction for follow-up of these patients [42].

Another important aspect of radiation toxicities in the cardiovascular system is the timeline for expected cardiac toxicities occurrence. A triggering point is the insidious side effects of radiation exposure since, in many cases of clinical entities, it takes many years from exposure until the clinical signs and symptoms exist. CAD, chronic pericarditis, pericardial constriction, and VHD may take several decades until their clinical presentation. At the same time, smaller time periods are needed for acute pericarditis, cardiomyopathies, and acute pericarditis to appear. Radiation dosage is an important consideration for cardiovascular toxicities. Since 30 Gy of radiation dosage is enough for cardiac consequences to existing and the usual radiation dosage for curative cancer treatment usually exceeds 35–40 Gy, it surely leads to serious dosage considerations involving cardiac toxicities.

A really important aspect of knowledge of the aforementioned mechanisms of cardiac toxicities and radiation dosage and techniques, as well as the timeline of cardiovascular toxicities, is the development of screening programs for early detection of toxicities and management of such adverse effects. Another important factor is that there is a novel area for the development of new risk factors for cardiac disease apart from classic cardiovascular risk factors and maybe new probability scores in cancer patient populations. Risk factors according to radiation treatment, such as the total dosage, the site of irradiation, or the cumulative dosage, are very important and might deteriorate the adverse cardiovascular effects. Furthermore, they may cause more severe disease, such as more central coronary lesions or greater extent or length of a stenosis, or even multivessel coronary artery disease or accelerate valvular disease progression or cause multivalvular disease. In addition, they can further increase fibrosis and lead to more severe heart failure or myocardiopathy and increase the risk of arrhythmic events. Risk scores for early detection of coronary artery disease should be considered taking into consideration all these factors related to cancer history or treatment dosages, whether radiation therapy or chemotherapy combinations were used, but also more precise information about treatment.

All the aforementioned will increase early detection of such toxicities, and early treatment options could be applied before irreversible tissue damage appear, which will increase the comorbidities and deteriorate the prognosis of such patients. Further knowledge of cardiovascular toxicities and the evolution of cardio-oncology will help in the advancement of care of these patients and early detection of disease through early screening and meticulous follow-up of patients for early detection of toxicities. Another important fact to be decided is how often follow-up evaluation of cancer patients should be conducted to detect preclinical structural heart disease and which modalities should be chosen for follow-up of cancer patients. Is simple transthoracic echocardiographic imaging enough, or multimodality imaging with CMR or nuclear cardiology will reveal more information on the pathophysiology of the disease? Newer cardiovascular imaging techniques, such as global longitudinal strain imaging, should be used to predict high-risk patients. Another important factor is whether biomarkers should be used and at which time point of the cancer treatment plan should be evaluated to help in the detection of high-risk toxicities in the cancer population even before structural heart disease appear on multimodality imaging. Brain natriuretic peptides and cardiac troponin are proposed as useful cardiac biomarkers for the screening of cancer patients. Maybe new cut-off values should be checked for cancer patients since cancer may increase the value of cardiac biomarkers apart from the detection of cardiac disease. In addition, there should be evaluated whether pharmacologic treatment and which agents can be used for the prevention of cardiovascular toxicities in irradiated patients.

## 4. Conclusions

Many pathophysiologic mechanisms are supposed to be responsible for the cardiovascular toxicities of radiation therapy. Microvascular and macrovascular complications can lead to the majority of cardiac adverse effects [43]. Many radiotherapy-associated risk factors are responsible such as the site of the radiation treatment beam (especially close proximity tissues such as the lung, breast, esophagus, and mediastinum), total dosage, the number of radiotherapy sessions, adjuvant chemotherapeutic agents used for cancer treatment but also patient individualized risk factors such as traditional cardiovascular risk factors, patient age, and gender, increase risk of cardiac toxicities. The evolution of cardiovascular and multimodality imaging techniques as well as biomarkers, are helping in the evaluation and early detection of cardiovascular toxicities in cancer patients population. The evolution of cardio-oncology as a new field of cardiovascular care is an important contribution. No matter all the advancements, much more knowledge is needed because many lacks of evidence in cancer populations generate questions about the follow-up of these patients.

## Figures and Tables

**Figure 1 ijms-24-06272-f001:**
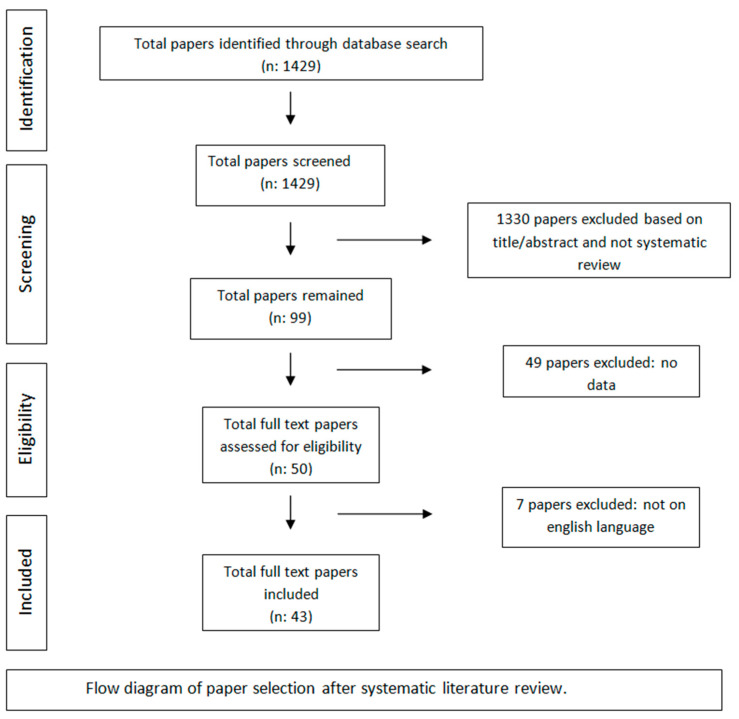
Flowchart of publication selection after systematic literature review.

**Figure 2 ijms-24-06272-f002:**
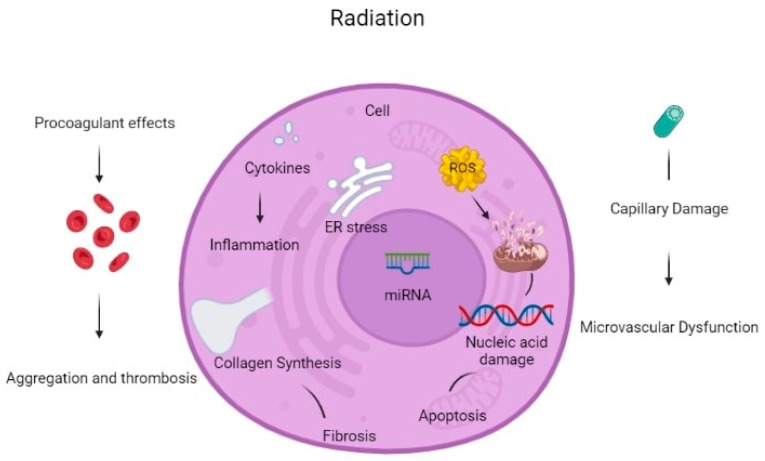
Summary of mechanisms of radiation-induced cardiac toxicities.

**Figure 3 ijms-24-06272-f003:**
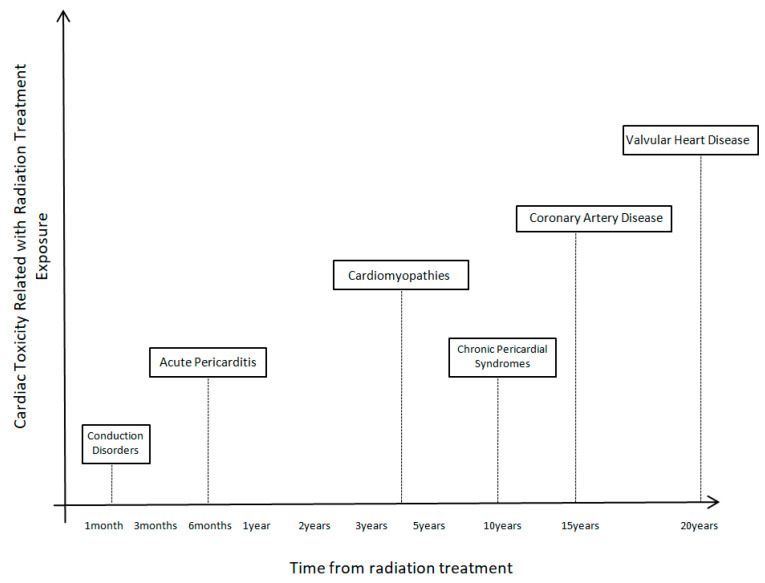
Time course of cardiac toxicities after radiation treatment.

**Table 1 ijms-24-06272-t001:** Molecular effects and clinical consequences of radiation therapy.

Molecular Effects of Radiation Exposure	Cardiovascular Toxicity
Endothelial Cell Injury, Microvascular Dysfunction, Accelerated atherosclerosis, Oxidative Stress, and Reactive Oxygen Species	Coronary Artery Disease
Pro-inflammatory mechanisms, Fibrosis, Cellular Infiltrations, miRNAs, DNA damage, Apoptosis	Heart Failure, Diastolic Dysfunction, Myocarditis, Cardiomyopathies
Fibrosis, Direct irradiation thickening, Calcifications	Valvular Heart Disease
Fibrosis, Direct damage/degeneration	Conduction Abnormalities and arrhythmias
Pro-inflammatory mechanisms	Pericardial Syndromes
Procoagulant effects	Arterial and venous thromboembolic events

**Table 2 ijms-24-06272-t002:** Cardiac toxicities after radiation therapy exposure and chemotherapeutic agents.

Cardiovascular System Adverse Effects of Cancer Treatments	Cardiac Toxicities of Radiation Therapy	Cardiac Toxicities of Chemotherapeutic Agents
Coronary Arteries	CAD	CAD (Fluoropyrimidines, VEGF inhibitors), Vasospasm, ACS (Fluoropyrimidines)
Valves	VHD	N/A
Heart Failure, Cardiomyopathies	HF	HF (Anthracyclines, Alkylating Agents, Proteasome Inhibitors, HER-2 antagonists, TKIs, VEGF, Taxanes)
Arrythmias	Conduction abnormalities	AF (Anthracyclines, Alkylating Agents, Fluoropyrimidines, TKIs, VEGF), Bradycardia (Taxanes)
Risk Factors	N/A	Hypertension (Proteasome Inhibitors)
Pericardial Syndromes	Acute and chronic	Acute pericarditis (Anthracyclines, Alkylating Agents), Chronic (ICI)
Thromboembolism	VTE, PE, Arterial	VTE, PE, Arterial (VEGF, Taxanes)
Myocardium	N/A	Myocarditis (ICI)
Pulmonary Vasculature	N/A	PH (TKIs)

ACS: Acute Coronary Syndromes, AF: Atrial Fibrillation, CAD: Coronary Artery Disease, ICI: Immune Checkpoint Inhibitors, HER: Human Epidermal Growth Receptor, HF: Heart Failure, N/A: Not Applicable, PE: Pulmonary Embolism, PH: Pulmonary Hypertension, TKIs: Tyrosine Kinase Inhibitors, VEGF: Vascular Endothelial Growth Factor, VHD: Valvular Heart Disease, VTE: Venous Thromboembolism.

## Data Availability

Not applicable.

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
