# Peer review of "Radiation Treatment Mechanisms of Cardiotoxicity: A Systematic Review"

_ijms, 2023, doi:10.3390/ijms24076272_

Round 1

Reviewer 1 Report

Cardiac toxicity is a known complication of radiotherapy in cancer treatments, in fact, cardiovascular disease is now the most common cause of mortality in cancer survivors. The authors aimed to summarize the mechanisms of radiation induced cardiac toxicities. Other excellent reviews have recently addressed this topic (doi: 10.7150/ijbs.35460) and the contribution of the present review is moderate.

 INTRODUCTION

The introduction is clear and concise.

 METHODS

My main concern is that the search strategy is limited to a single database. Why were other databases besides PubMed not used?

 RESULTS

Some important reviews on the subject have not been detected in the search process:

            A 2019 review on mechanism, prevention and classification of cardiac toxicity of radiotherapy: doi: 10.7150/ijbs.35460

A Systematic Review available on line since May 6, 2022 about radiation-induced cardiotoxicity in model animals should be included: DOI: 10.1016/j.radonc.2022.04.030

Some important mechanisms of radiation-induced cardiac toxicity, such as ER stress and miRNAs, are missing from Figure 2.

Finally, in general, reading could be eased considerably. The manuscript requires moderate English changes, for instance:

Pag5, line 104 … causing increase in production or Reactive Oxygen Species…

Page 6, line 120 …Oxidative stress increase TGF-β1 increase…

Reviewer 2 Report

the subject matter presented by the authors of the manuscript is extremely important in many areas of clinical medicine. The consequences and, if possible, prevention are very important - but this is difficult to obtain, so early diagnosis and quick treatment come to the fore.
An interesting manuscript, however, I propose an introduction:
- a table comparing the effects of cardiotoxicity after radiotherapy and chemotherapy
- a table linking molecular effects with clinical consequences
- echocardiography is an examination that should be performed routinely before radiotherapy and sequentially during it - please refer to the schemes proposed in the latest ESC recommendations
- please discuss the most typical changes in the ECHO examination
- CAD and angio diagnostics - please discuss the most typical locations and nature of the lesions. This is an element affecting, for example, the need to perform treatments in properly equipped catlabs (IVUS, OCT, rotablation, shock wave baloons, cutting balloons, etc...). A different approach to fibrotic lesions is different to hard calcified lesions
- the authors treated MR diagnostics as if marginally ... Please elaborate.
- what is the approach to patients after previous coronary interventions - imaging of changes inside stents is rather unreliable...
- it would be good to include a chapter on prevention options. pharmacotherapy?
- please address the risk of thromboembolic events, atrial fibrillation. Overall, these patients often require monitored anticoagulant therapy. In this situation, please pay attention to interactions and the possibility of monitoring (e.g. inhibition of fibrosis with nintedanib and the use of anticoagulants, DOAC monitored therapy, etc.)

Round 2

Reviewer 2 Report

I believe that the revised manuscript may be considered for publication. personally I would prefer to use abbr DOAC (direct oral anticoagulant) instead of NOAC (they are not that new anymore...)... but it's a matter of choice...